# Translational gene expression control in *Chlamydia trachomatis*

**Nicole A. Grieshaber, Travis J. Chiarelli, Cody R. Appa, Grace Neiswanger, Kristina Peretti, Scott S. Grieshaber** *

Department of Biological Sciences, University of Idaho, Moscow, ID, United States of America

* scottg@uidaho.edu

## Abstract

The human pathogen *Chlamydia trachomatis* proceeds through a multi phenotypic developmental cycle with each cell form specialized for different roles in pathogenesis. Understanding the mechanisms regulating this complex cycle has historically been hampered by limited genetic tools. In an effort to address this issue, we developed a translational control system to regulate gene expression in *Chlamydia* using a synthetic riboswitch. Here we demonstrate that translational control via a riboswitch can be used in combination with a wide range of promoters in *C. trachomatis*. The synthetic riboswitch E, inducible with theophylline, was used to replace the ribosome binding site of the synthetic promoter T5-lac, the native chlamydial promoter of the *pgp4* plasmid gene and an anhydrotetracycline responsive promoter. In all cases the riboswitch inhibited translation, and high levels of protein expression was induced with theophylline. Combining the Tet transcriptional inducible promoter with the translational control of the riboswitch resulted in strong repression and allowed for the cloning and expression of the potent chlamydial regulatory protein, HctB. The ability to control the timing and strength of gene expression independently from promoter specificity is a new and important tool for studying chlamydial regulatory and virulence genes.

## Introduction

The bacterial species *Chlamydia trachomatis* (*Ctr*), are a group of human pathogens composed of over 15 distinct serovars causing trachoma, the leading cause of preventable blindness, and sexually acquired infections of the urogenital tract. According to the CDC, *Ctr* is the most frequently reported sexually transmitted infection in the United States, costing the American healthcare system nearly $2.4 billion annually [1, 2]. These infections are widespread among all age groups and ethnic demographics, infecting ~3% of the human population worldwide [3]. In women, untreated genital infections can result in devastating consequences such as pelvic inflammatory disease, ectopic pregnancy, and infertility [4, 5]. Every year, there are over 4 million new cases of *Ctr* in the United States [6, 7] and an estimated 152 million cases worldwide [8]. Understanding the genetic factors that mediate infection and disease has historically been hindered by the lack of good genetic tools. This has changed dramatically in the last few years with advances in chlamydial transformation. The ability to introduce genetically

**Data Availability Statement:** RNA-seq data is available from the NCBI's Sequence Read Archive (SRA) submission number PRJNA756251. All other relevant data are within the paper and its Supporting Information files.

**Funding:** This work was supported by National Institutes of Health Granst R01AI130072, R21AI135691 and R21AI113617 (NG, SG). The funders had no role in study design, data collection and analysis, decision to publish, or preparation of the manuscript.

**Competing interests:** The authors have declared that no competing interests exist.

manipulatable plasmids into *Ctr* has created multiple opportunities to bring genetic manipulation techniques to the field [9–11]. The ability to alter the expression levels and timing of proteins involved in chlamydial pathogenesis is an important tool in teasing apart the mechanisms that control chlamydial infections.

Here we demonstrate the use of an inducible translational control system using a synthetic riboswitch in *Ctr*. Riboswitches are naturally occurring mRNA elements that regulate gene expression in all domains of life [12]. In bacteria, riboswitches generally function to interfere with translation of the mRNA. Riboswitches contain an aptamer sequence that binds a cognate ligand causing the mRNA to adopt an alternative secondary-structure conformation. In bacteria, the changes in mRNA secondary structure can control the availability of the ribosome binding site on the mRNA. A collection of synthetic riboswitches was recently developed through screening and rational design [13]. These riboswitches respond to theophylline, a caffeine analog, and function through a translation initiation mechanism. We successfully adapted one of these theophylline inducible riboswitches, termed E riboswitch, to control gene expression in *Ctr*. Additionally, we demonstrated that translational control can be used in conjunction with constitutive promoters, inducible promoters and native chlamydial promoters, demonstrating the versatility of translational inducible control of gene expression in a variety of use cases.

## Material and methods

### Cell culture

Cell lines were obtained from the American Type Culture Collection. Cos-7 cells (CRL-1651) were grown in RPMI-1640, supplemented with 10% FBS and 10 μg/mL gentamicin (Cellgro). *Chlamydia trachomatis* serovar L2 (LGV Bu434) was grown in Cos-7 cells. Elementary Bodies (EBs) were purified by density gradient (DG) centrifugation essentially as described [14] following 43–45 h of infection. EBs were stored at -80°C in Sucrose Phosphate Glutamate (SPG) buffer (10 mM sodium phosphate [8mM K2HPO4, 2mM KH2PO4], 220 mM sucrose, 0.50 mM l-glutamic acid, pH 7.4) until use.

### Vector construction

All *Ctr* expression constructs used p2TK2-SW2 [15] as the backbone and cloning was performed using the In-fusion HD EcoDry Cloning kit (FisherScientific). Primers and geneblocks (gBlocks) were ordered from Integrated DNA Technologies (IDT) and are noted in S1 Table. All constructs are penicillin (*bla*) resistant except p2TK2-SW2-euoprom-ngLVA which is spectinomycin (*aadA*) resistant.

**p2TK2-SW2-T5-E-clover-3xflag.** An E-clover-3xFlag fragment was ordered as a gBlock and inserted between the T5 promoter and IncD terminator of p2TK2-SW2 to generate p2TK2-SW2-E-clover-3xFlag. The backbone was generated using primers 5' E-clover-Flag bb and 3' E-clover-Flag bb.

**p2TK2-SW2-E-hctB-3xFlag.** The *hctB* ORF was amplified from *Ctr* L2(434) using the primers indicated in S1 Table. The fragment was used to replace Clover in p2TK2-SW2- E-clover-3xFlag. The primers used to generate the backbone are described in S1 Table.

**p2TK2-SW2-Tet-J-E-clover-3xflag.** A gBlock encoding the Tet repressor, Tet promoter and the riboJ ribozyme insulator (S1 Table) was inserted upstream of the E riboswitch of p2TK2-SW2-E-clover-3xFlag, replacing the T5 promoter.

**p2TK2-SW2-Tet-J-E-hctB-3xFlag.** The *hctB* ORF was amplified from *Ctr* L2(434) using the primers 5' Tet-J-HctBi and 3' Tet-J-HctBi. The fragment was used to replace Clover in p2TK2-SW2 -Tet-J-E-clover-3xFlag.

**p2TK2-SW2-nprom-E-pgp4-3xFlag.** An E-pgp4-3xFlag fragment was ordered as a gBlock and inserted between the pgp4 native promoter and the IncD terminator of p2TK2-SW2. The backbone was generated using the primers indicated in S1 Table.

**p2TK2-SW2-T5-E-ngLVA-3xFlag and p2TK2-SW2-Tet-J-E-ngLVA-3xFlag.** A neon-greenLVA (ngLVA) fragment was ordered as a gBlock from IDT and inserted to replace Clover of both p2TK2-SW2-E-clover-3xFlag and p2TK2-SW2-Tet-J-E-clover-3xflag. The primers indicated in S1 Table were used for both plasmids to generate the back bone.

**p2TK2-SW2-euoprom-ngLVA.** The primers 5' ngLVAi and 3' ngLVAi (S1 Table) were used to amplify the ngLVA fragment from E-ngLVA-3xFlag and inserted to replace Clover of p2TK2-SW2-euoprom-Clover (*aadA*) described by Chiarelli et al. [16]. The primers indicated in S1 Table were used to generate the back bone.

## Chlamydial transformation and isolation

Transformation of Ctr L2 was performed essentially as previously described [17]. Briefly, $1x10^8$ EBs + >2µg DNA/well were used to infect a 6 well plate. Transformants were selected over successive passages with 1U/ml penicillin G or 500µg/ml spectinomycin as appropriate for each plasmid. The new strain was clonally isolated via successive rounds of inclusion isolation (MOI, <1) using a micromanipulator. Clonality of each strain was confirmed by isolating the plasmid, transforming into E. coli and sequencing six transformants.

## Fluorescence staining

Cos7 cells on coverslips were infected with the indicated strains. Protein expression regulated by the E-riboswitch only was induced at 16 hpi with 0.5mM theophylline (dissolved in RPMI media) (Acros Organics, Thermo Scientific™). Protein expression regulated by both the Tet promoter and the E-riboswitch were induced at 16 hpi with 0.5mM theophylline and 30ng/ml anhydroTetracycline (Acros Organics, Thermo Scientific™). Theophylline (theo) was dissolved in RPMI media to a concentration of 50 mM and diluted 1:100 to induce protein expression. AnhydroTetracycline (aTc) was dissolved in DMSO to 10mg/ml and diluted to 30ng/ml in RPMI for protein expression. DMSO diluted 1:333,333 in RPMI served as vehicle control when appropriate. Samples were fixed with 4% buffered paraformaldehyde at 24 hpi and stained with Monoclonal anti-Flag M2 antibody (1:500, Sigma, Thermo Scientific™) and alexa 488 anti-mouse secondary antibody to visualize expressing Chlamydia. DAPI was used to visualize DNA. Coverslips were mounted on a microscope slide with a MOWIOL® mounting solution (100 mg/mL MOWIOL® 4–88, 25% glycerol, 0.1 M Tris pH 8.5).

Fluorescence images were acquired using a Nikon spinning disk confocal system with a 60x oil-immersion objective, equipped with an Andor Ixon EMCCD camera, under the control of the Nikon elements software. Images were processed using the image analysis software ImageJ (http://rsb.info.nih.gov/ij/). Representative confocal micrographs displayed in the figures are maximal intensity projections of the 3D data sets, unless otherwise noted.

## Live cell imaging

Infected monolayers of Cos7 cells grown in a glass bottom 24 well plate were induced at 16 hpi with either 0.5mM theophylline only or the indicated concentrations of theophylline and anhydroTetracycline. Plates were imaged immediately upon induction.

Live cell imaging was achieved using an automated Nikon epifluorescent microscope equipped with an Okolab (http://www.oko-lab.com/live-cell-imaging) temperature controlled stage and an Andor Zyla sCMOS camera (http://www.andor.com). Images were taken every fifteen minutes for a further 36 hours. Multiple fields of view of multiple wells were imaged.

The fluorescence intensity of each inclusion over time was tracked using the ImageJ plugin Trakmate [18]. and the results were averaged and plotted using python and matplotlib [19].

## Replating assay

*Ctr* were isolated by scraping the infected monolayer into media and pelleting at 17200 rcfs. The EB pellets were resuspended in RPMI via sonication and seeded onto fresh monolayers in a 96-well microplate in a 2-fold dilution series. Infected plates were incubated for 24 hours prior to fixation with methanol and stained with DAPI and *Ctr* MOMP Polyclonal Antibody, FITC (Fishersci). The DAPI stain was used for automated microscope focus and visualization of host-cell nuclei and the anti-*Ctr* antibody for visualization of EBs and inclusion counts. Inclusions were imaged using a Nikon Eclipse TE300 inverted microscope utilizing a scopeLED lamp at 470nm and 390nm, and BrightLine band pass emissions filters at 514/30nm and 434/17nm. Image acquisition was performed using an Andor Zyla sCMOS in conjunction with µManager software. Images were analyzed using ImageJ software and custom scripts. Statistical comparisons between treatments were performed using an ANOVA test followed by Tukey's Honest Significant Difference test.

## Western analysis

Infected monolayers were lysed in reducing lane marker sample buffer and protein lysates were separated on 12% SDS-PAGE gels and transferred to a Nitrocellulose Membrane for western analysis of the Flag-tagged protein or ß-tubulin I as a loading control. The membrane was blocked with PBS + 0.1% Tween 20 (PBS-T) and 5% nonfat milk prior to incubating in either monoclonal anti-Flag M2 antibody (1:40,000, Sigma, Thermo Scientific™) or anti-beta I Tubulin, Clone: SAP.4G5 (1:20,000, Novus Biologicals™, Fishersci) overnight at 4˚C followed by Goat-anti Mouse IgG-HRP secondary antibody (Invitrogen™) at room temperature for 2 hours. The membrane was developed with the Supersignal West Dura luminol and peroxide solution (Thermo Scientific™) and imaged using an Amersham Imager 600.

## Glycogen staining

Monolayers were infected with the indicated strains and induced with 0.5mM theophylline at the time of infection. At 36 hpi, the media was removed and the samples were stained with 1 ml of a 1:50 dilution of 5% iodine stain (5% potassium iodide and 5% iodine in 50% ethanol) in PBS for 10 min. Samples were then stained in 1:50 Lugol's iodine solution in PBS (10% potassium iodide and 5% iodine in ddH2O) and imaged directly. Images were acquired using a Nikon microscope using phase brightfield illumination and an Andor Zyla sCMOS camera.

## RNA-Seq

Total RNA was isolated from cells infected with L2-Tet-J-E-hctB-flag. Expression of HctB was induced with 0.5 mM theophylline and 30ng/ml aTc at 15 hpi and the *Ctr* isolated at 24 hpi on ice. Briefly, the infected monolayer was scraped into ice cold PBS, lysed using a Dounce homogenizer and the *Ctr* isolated over a 30% MD-76R pad. Total RNA was isolated using TRIzol reagent (Life Technologies) following the protocol provided and genomic DNA removed (TURBO DNA-free Kit, Invitrogen). The enriched RNA samples were quantified and the libraries built and barcoded by the IBEST Genomics Resources Core at the University of Idaho. The libraries were sequenced by University of Oregon sequencing core using the Illumina NovaSeq platform. RNA-seq reads were aligned to the published *C. trachomatis* L2 Bu 434 genome using the bowtie2 aligner software [20]. Reads were quantified using HTseq [21]

Statistical analysis and normalization of read counts was accomplished using DESeq2 in R [22]. Log2fold change and statistics were also calculated using DESeq2. Heatmaps and hierarchical clustering were generated and visualized using python with pandas and the seaborn visualization package [23]. Aligned reads are accessible from the NCBI's Sequence Read Archive (SRA) submission number PRJNA756251.

## Results

### Translational control of gene expression from a synthetic constitutive promoter

Controlling the timing and level of gene expression is an important tool for uncovering the function of genes that are involved in chlamydial pathogenesis. We developed an inducible expression system for use in *C. trachomatis* using a synthetic riboswitch that binds the small molecule theophylline [23, 24]. We used the synthetic riboswitch E behind a T5-lac promoter (T5) to drive expression of the GFP variant, Clover [25, 26] (Fig 1A). The T5-lac promoter is a hybrid promoter made from the phage T5 early promoter and the lac-operon [27]. The E riboswitch when not bound to theophylline folds to block the initiation of translation [28]. However, when the riboswitch binds theophylline the ribosome binding site is no longer obscured by the RNA secondary structure allowing for efficient translation. A T5-E-clover-3xFlag fragment was cloned into the chlamydial plasmid p2TK2-SW2 [15, 29] to make the p2TK2-SW2-T5-E-clover-3xflag plasmid (Fig 1A) and transformed into *Ctr* L2 resulting in the strain L2-E-clover-flag. Cos-7 cells were infected with these transformants and Clover expression was evaluated by western blotting. Cells were infected and treated with either theophylline or vehicle at 16 hours post infection (hpi) and cell lysates were analyzed for protein production at 30 hpi. Clover expression was tightly regulated and was only detectable in the theophylline treated sample (Fig1B). In addition to western blotting we also evaluated the expression of Clover using confocal microscopy. Cos-7 cells grown on coverslips were infected with the L2-E-clover-flag strain and Clover expression was induced with theophylline at 16 hpi. The coverslips were fixed at 30 hpi and imaged for Clover expression using confocal microscopy. Again, only inclusions treated with theophylline had fluorescent *Ctr* (Fig 1C).

To determine the effects of theophylline and gene expression induction on chlamydial growth dynamics the production of infectious EBs using a reinfection inclusion forming unit assay was performed. Cos-7 cells were infected with L2-E-clover-flag and induced with theophylline at 16 hpi. EBs were harvested at 30 hpi and 48 hpi. Clover induction with theophylline had no significant effect on EB production at 30 hpi (S1 Fig) or 48 hpi (Fig 1D). The control of expression of ectopic proteins to assess their function in pathogenesis needs to be highly customizable as too little or too high concentrations may mask the phenotype of interest. Therefore we assessed the dose responsiveness of the E riboswitch to its ligand theophylline. Gene expression was measured using live-cell time-lapse microscopy and particle tracking to quantify the fluorescent expression of individual inclusions over time [19, 30]. This technique allows for the tracking of gene expression in multiple individual inclusions over the entire developmental cycle. Cos-7 cells were plated in a glass bottom 24 well plate and infected with L2-E-clover-flag at an multiplicity of infection (MOI) of ~0.5. At 16 hpi theophylline at 1mM, 0.5 mM, 0.25mM, 0.0125 mM, 0.00625 mM, and 0.00312 mM was added to individual wells to induce Clover expression; images were taken every 15 minutes for 48 hours. Clover expression followed a dose response with almost immediate detection of fluorescence with 1mM theophylline and a delayed response at the lowest dose, 0.00312 mM (Fig 1E). The response increased through the life of the inclusion; this increase overtime also followed a dose response (Fig 1E).

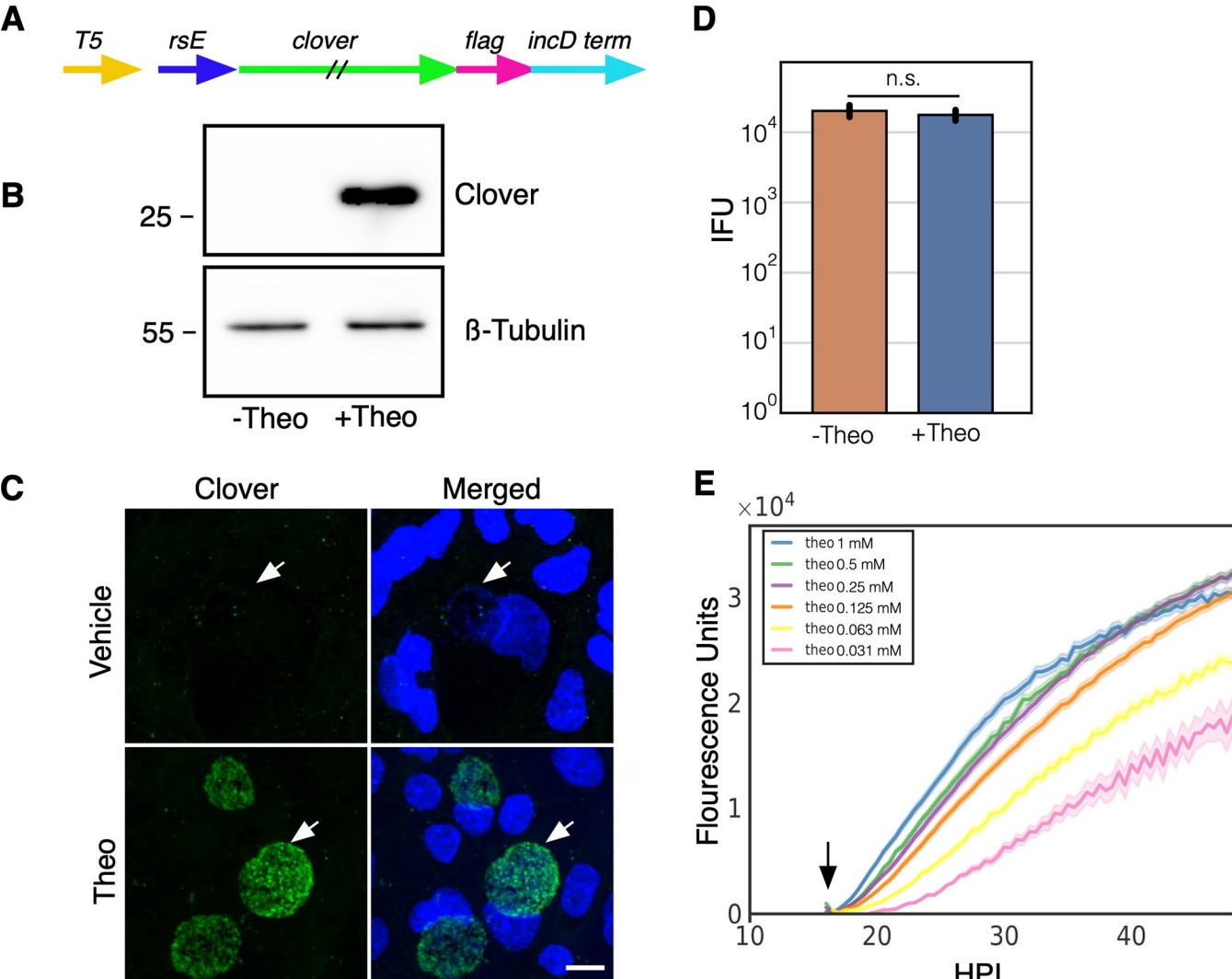

**Fig 1. Characterization of p2TK2-SW2-T5-E-clover-flag.** A) Schematic of the E-clover expression construct consisting of the T5-lac promoter (T5), riboE riboswitch (rsE) and the ORF for the clover fluorescent protein with an inframe 3x flag tag. B) Anti-flag western blot of Cos-7 cells infected with L2-T5-E-clover-flag comparing expression of theophylline treated and untreated cultures. Cells were induced or not with 0.5 mM theophylline at 16 hpi and proteins were harvested at 30 hpi. A ß-tubulin I western blot served as a sample loading control. C) Confocal micrographs of Cos-7 cells infected with L2-T5-E-clover-flag and induced or not with 0.5 mM theophylline at 16 hpi and fixed and stained with DAPI for microscopy at 30 hpi. DAPI (blue), Clover (green). Arrow indicates the position of the chlamydial inclusion. Size bar = 10 μm. D) Cos-7 cells were infected with L2-T5-E-clover-flag and the production of infectious progeny was determined at 48 hpi after 0.5 mM theophylline induction or vehicle only. E) Cos-7 cells were infected with L2-T5-E-clover-flag, treated with varying dilutions of theophylline at 16 hpi (1 mM, 0.5 mM, 0.25 mM, 0.125 mM, 0.0625 mM, 0.03125mM) and imaged for 50 hours using live cell imaging. The Clover expression intensities from >50 individual inclusions were monitored via automated live-cell fluorescence microscopy and average intensities were plotted. Live cell imaging demonstrated that Clover induction was dose responsive. Cloud represents SEM. Y-axes are denoted in scientific notation. Error bars = SEM. n. s. denotes p-values > 0.05.

## Translational control of gene expression from a native chlamydial promoter

The use of non endogenous promoters for ectopic expression is an important tool for understanding protein function. However, these systems lack the ability to control gene expression through native gene regulation making it difficult to modulate expression at biologically relevant times or in the correct cell subspecies. This is especially true for *Ctr* as it proceeds through a time dependent developmental cycle that includes multiple phenotypic cell types. Therefore,

the use of translational control was tested in concert with a native chlamydial promoter. We tested the effectiveness of translational control on the pgp4 native plasmid gene. Pgp4 is a regulator of other plasmid genes as well as chromosomal genes [31–33]. *Ctr* strains with pgp4 knocked out from the native plasmid show marked changes in gene expression and a phenotypic loss of glycogen accumulation [33]. To assess the ability to regulate translation of transcripts from a native promoter, the E riboswitch was cloned upstream of the pgp4 open reading frame (ORF) replacing the predicted ribosome binding site (Fig 2A). The insertion was designed to not disrupt the native promoter region of pgp4. A flag tag was also added in frame to the end of the pgp4 ORF creating the plasmid p2TK2-SW2-nprom-E-pgp4-3xflag (Fig 2A). This plasmid was then transformed into *Ctr* L2 to create L2-nprom-E-pgp4-flag. To assess expression, Cos-7 cells were infected with the L2-nprom-E-pgp4-flag strain in the presence of 0.5 mM theophylline. Expression was assessed by western blotting and flag tag detection was greatly increased in theophylline treated samples (Fig 2B). The control of pgp4 expression was also assessed by confocal microscopy. Cos-7 cells were plated on coverslips and infected with L2-nprom-E-pgp4-flag and treated with 0.5 mM theophylline at infection. The coverslips were fixed at 30 hpi, stained with an anti-flag antibody and DAPI for visualization (Fig 2C). Like for the western blotting experiment, flag epitope detection was dramatically increased in the theophylline induced samples (Fig 2C). The effect of modulating pgp4 expression on EB production was determined using a re-infection assay. Cos-7 cells were infected with L2-nprom-E-pgp4-flag and translation was induced at infection with 0.5 mM theophylline. EBs were harvested at 48 hours and monolayers were re-infected and inclusions quantified. Repression of pgp4 expression resulted in a slight but statistically significant increase in infectious progeny as compared to induced pgp4 expression (Fig 2D). Pgp4 positively regulates the expression of GlgA which is involved in accumulation of glycogen in the inclusion. When pgp4 expression is missing the *Ctr* inclusion does not accumulate glycogen and is phenotypically similar to the plasmidless L2 strain, L2R [33]. Therefore, we tested the ability of translational regulation to control glycogen accumulation in the inclusion. Cos-7 cells were infected with the L2-nprom-E-pgp4-flag strain and treated or not with 0.5 mM theophylline at the time of infection. Cells were stained for glycogen accumulation at 36 hpi using Lugol's iodine solution as previously described [33]. As expected from the flag detection of expression, glycogen staining was strongly detected in inclusions that were treated with theophylline (Fig 2E). The uninduced inclusions were morphology similar to inclusions formed by the plasmidless strain L2R which lack glycogen accumulation (Fig 2E).

## Transcriptional and translational control of gene expression

The E riboswitch partnered with either the T5 promoter or native pgp4 promoter offered very tight expression control. There was no detectable Clover or Pgp4 by western blotting and no fluorescence from Clover or Flag staining detected using confocal microscopy (Figs 1 and 2). However, we attempted to use the T5-E system to ectopically express the *Ctr* histone like protein HctB. Clover was replaced on the p2TK2-SW2-T5-E-clover-3xflag plasmid with HctB creating p2TK2-SW2-T5-E-hctB-3xflag. This construct was then transformed into *Ctr*. Although we successfully isolated transformants, when the plasmids were purified and sequenced, the promoter region of the plasmid was mutated in every case. We reasoned that the HctB protein expression was leaky enough to lead to small amounts of HctB accumulation despite the translation inhibition of the E riboswitch, thereby inhibiting the chlamydial developmental cycle. We therefore sought to create an extremely tightly regulated inducible expression system by combining inducible transcription with inducible translation. For this construct we added the Tet repressor and replaced the T5 promoter with a Tet promoter containing Tet operator sites

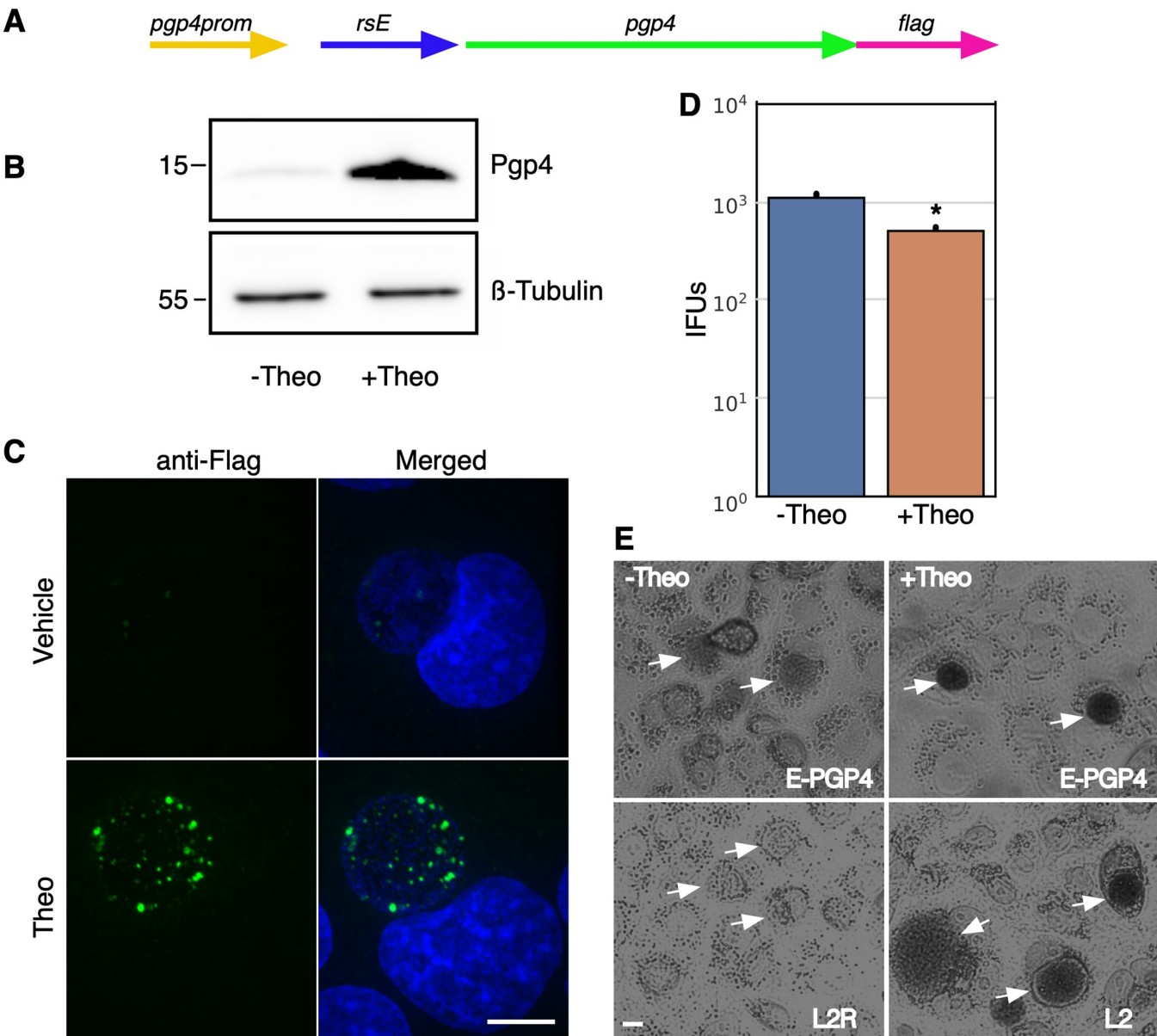

**Fig 2. Characterization of p2TK2-SW2-nprom-E-pgp4-flag.** A) Schematic of the nprom-E-pgp4-flag construct consisting of the native pgp4 promoter, the riboE riboswitch (rsE), and the pgp4 ORF with an inframe 3x flag tag. B) Anti-flag western blot of Cos-7 cells infected with L2-nprom-E-pgp4-flag comparing expression of theophylline treated and untreated cultures. Cells were induced or not with 0.5mM theophylline at 16 hpi and proteins were harvested at 30 hpi. A ß-tubulin I western blot served as a loading control. C) Confocal micrographs of Cos-7 cells infected with L2-nprom-E-pgp4-flag, induced or not with 0.5 mM theophylline at 16 hpi and fixed and stained with DAPI to detect DNA. The flag tag was detected using a primary antibody to the tag and an alexa 488 anti-mouse secondary antibody (green). Size bar = 10 μm. D) Cos-7 cells were infected with L2-nprom-E-pgp4-flag and the production of infectious progeny was determined at 48 hpi after 0.5 mM theophylline induction or vehicle only. E) Iodine staining of glycogen in the inclusion of Cos-7 cells infected with L2-nprom-E-pgp4-flag after 0.5 mM theophylline induction at 16 hpi or vehicle only. Arrows indicate the location of the chlamydial inclusions. Asterisk denotes p-value < 0.05. Error bars = SEM.

in the p2TK2-SW2-T5-E-clover-3xflag plasmid [34] (Fig 3A). In addition to replacing the T5 promoter with the Tet promoter, a ribozyme insulator was added to the E riboswitch to decouple the promoter from the riboswitch (Fig 3A). The riboJ ribozyme insulator is a self cleaving 75 nucleotide sequence from the satellite RNA of tobacco ringspot virus (sTRSV) followed by a 23 nucleotide hairpin [35]. After transcription, the ribozyme self-cleaved, removing

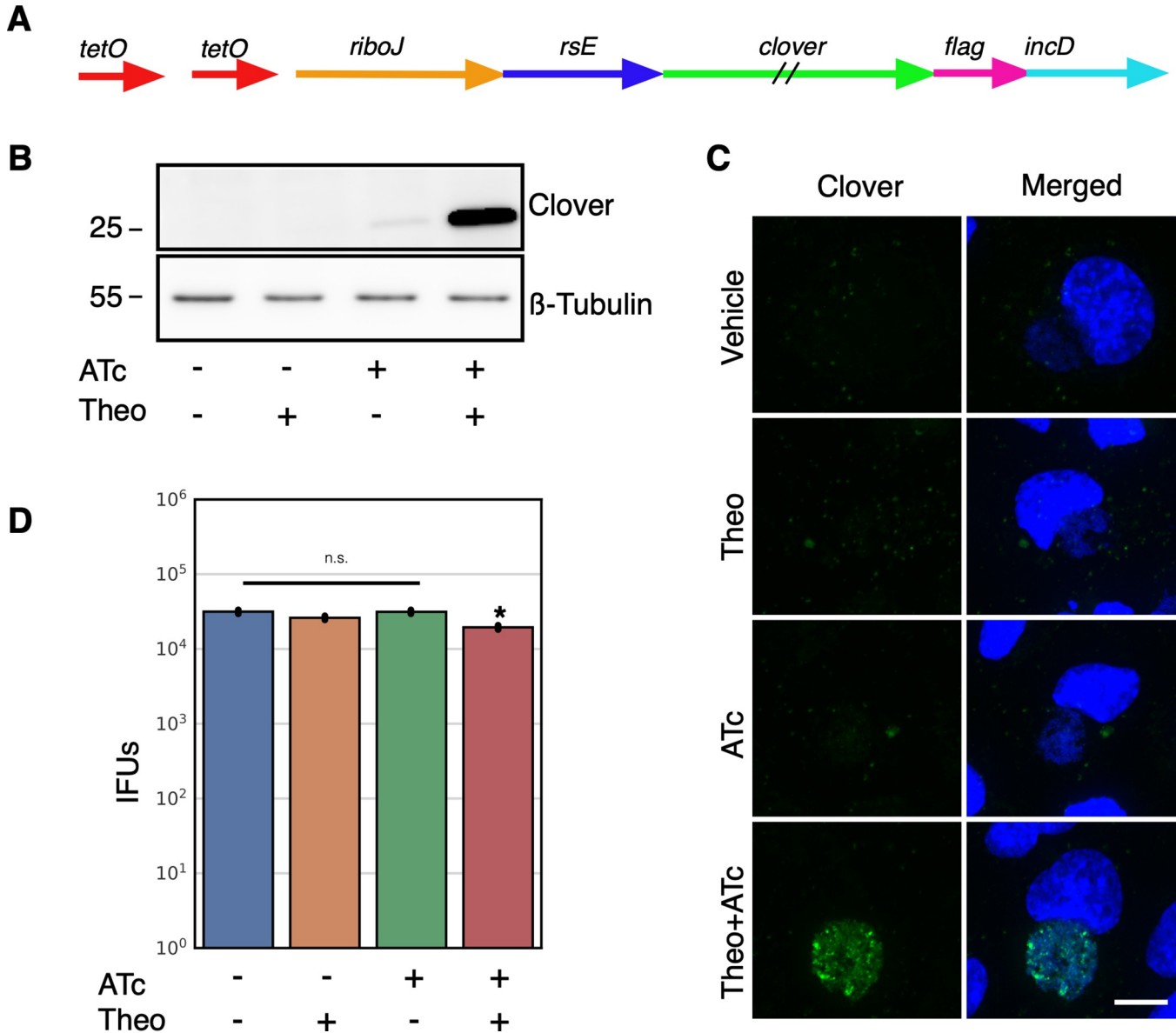

**Fig 3. Characterization of p2TK2-SW2-Tetp-riboJ-E-clover-flag.** A) Schematic of the Tet-riboJ-E-clover-flag construct consisting of the tet repressor, tet promoter, riboJ insulator, riboE riboswitch (rsE) and the ORF for the clover fluorescent protein containing an inframe 3x flag tag. B) Anti-flag western blot of Cos-7 cells infected with L2-Tet-J-E-clover-flag comparing expression of theophylline treated and untreated cultures. Cells were induced with 0.5 mM theophylline, 30ng/ml aTc, both aTc and theophylline or vehicle only at 16 hpi and proteins were harvested at 30 hpi. An anti-ß-tubulin I western blot served as a loading control. C) Confocal micrographs of Cos-7 cells infected with L2-Tet-J-E-clover-flag, induced with 0.5 mM theophylline, 30ng/ml aTc, both aTc and theophylline or vehicle only at 16 hpi and fixed and stained with DAPI (blue) for confocal microscopy at 30 hpi. Clover expression (green) was evident in cells treated with aTc and Theophylline. Size bar = 10 μm. D) Cos-7 cells were infected with L2-Tet-J-E-clover-flag and the production of infectious progeny was determined at 48 hpi after induction with 0.5 mM theophylline, 30ng/ml aTc, both aTc and theophylline or vehicle only at 16 hpi. Production of infectious progeny was determined using a reinfection assay. Asterisk denotes p-value < 0.05. Error bars = SEM.

upstream sequences, eliminating the promoter-associated RNA leader (S2 Fig). This resulted in transcripts with a small hairpin region just upstream of the E riboswitch that we hypothesized would not affect the aptamer function of the riboswitch. We used this same regulatory scheme to control the expression of both Clover and HctB resulting in plasmids p2TK2-SW2-Tet-J-E-clover-3xflag and p2TK2-SW2-Tet-J-E-hctB-3xflag. For the HctB clone

we used the AUG start site and the first three codons of the Clover gene followed by the HctB ORF without the AUG. We chose to use the first three codons of Clover as some genes in *Ctr* have small RNA regulatory sites at the beginning of the gene and wanted to avoid any native regulation [36]. These constructs were transformed into *Ctr* creating the strains L2-Tet-J-E-clover-flag and L2-Tet-J-E-hctB-flag.

Expression of Clover from p2TK2-SW2-Tet-J-E-clover-3xflag was evaluated by western blotting. Cos-7 cells were infected with the L2-Tet-J-E-clover-flag strain and expression was induced with the addition of either anhydroTetracycline (aTc) 30ng/ml or theophylline 0.5 mM alone, or both combined at 16 hpi. At 30 hpi protein from the infected cells was harvested, separated by PAGE and blotted to nitrocellulose for detection. Clover expression was detected using an anti-flag antibody. As expected Clover expression was only detected in samples induced with both aTc and theophylline (Fig 3B). We also evaluated Clover expression using confocal microscopy. Cos-7 cells plated on glass coverslips were infected for 16 hours before induction of expression with aTc 30ng/ml, theophylline 0.5 mM or both combined. The coverslips were fixed at 30 hpi, stained with DAPI and visualized by confocal microscopy. Again, robust Clover expression was only evident when both transcription and translation were induced (Fig 3C).

The effects of induction of this system was evaluated for effects on the chlamydial developmental cycle. The impact of induction on the production of infectious EBs was measured using an inclusion forming reinfection assay (IFU). Cos-7 cells were infected with L2-Tet-J-E-clover-flag and Clover expression was induced with aTc 30ng/ml, theophylline 0.5 mM or both combined at 16 hpi. EBs were harvested at both 30 hpi and 48 hpi to evaluate the production of infectious progeny (Fig 3D and S1 Fig). Each inducer alone had no effect on IFU formation. However, the addition of both inducers had a very small but statistically significant reduction in IFUs suggesting the expression of Clover resulted in a slight impact to the chlamydial developmental cycle (Fig 3D and S1 Fig).

To assess the effects of the induction of transcription or translation order we measured the kinetics of Clover expression using live cell imaging as described earlier. Cos-7 cells plated into 24 well glass bottom plates were infected with L2-Tet-J-E-clover-flag and either treated with aTc (30ng/ml) at infection and treated with a decreasing dose of theophylline (2 fold dilutions from 1 mM to 0.0312 mM) at 16 hpi (Fig 4A) or theophylline (0.5 mM) at infection followed by a decreasing dose of aTc (2 fold dilutions from 60 ng/ml to 1.25 ng/ml) at 16 hpi (Fig 4B). Infected cells were imaged for Clover expression every 30 minutes for 50 hours. Gene expression was quantified using live-cell time-lapse microscopy and particle tracking to quantify the fluorescent expression of individual inclusions over time [19]. Clover expression using transcriptional induction followed by translational induction demonstrated a robust dose response. Expression was detectable almost immediately after theophylline addition and detected at the lowest dose of 0.0312 mM theophylline (Fig 4A). Interestingly, max expression kinetics was observed with 0.5 mM of theophylline while 1 mM resulted in less expression suggesting potential toxicity at high concentrations (Fig 4A). When transcription was induced first (aTc) followed by translational induction at 16 hpi, expression was again initiated with little delay and a very strong dose response was observed (Fig 4B). Transcriptional induction with aTc resulted in higher expression as we did not reach a point of toxicity. This resulted in the highest expression being at the highest concentration of aTc (60 ng/ml) (Fig 4B). No induction was observed at the lowest aTc concentration (1.25 ng/ml). Notably, transcriptional induction followed by translational induction resulted in slightly higher induction as compared to translational induction followed by transcriptional induction as can be seen by comparing aTc 30 ng/ml followed by 0.5 mM theophylline at 16 hpi to 0.5 mM theophylline at infection followed by 30 ng/ml of aTc at 16 hpi (Fig 4A and 4B).

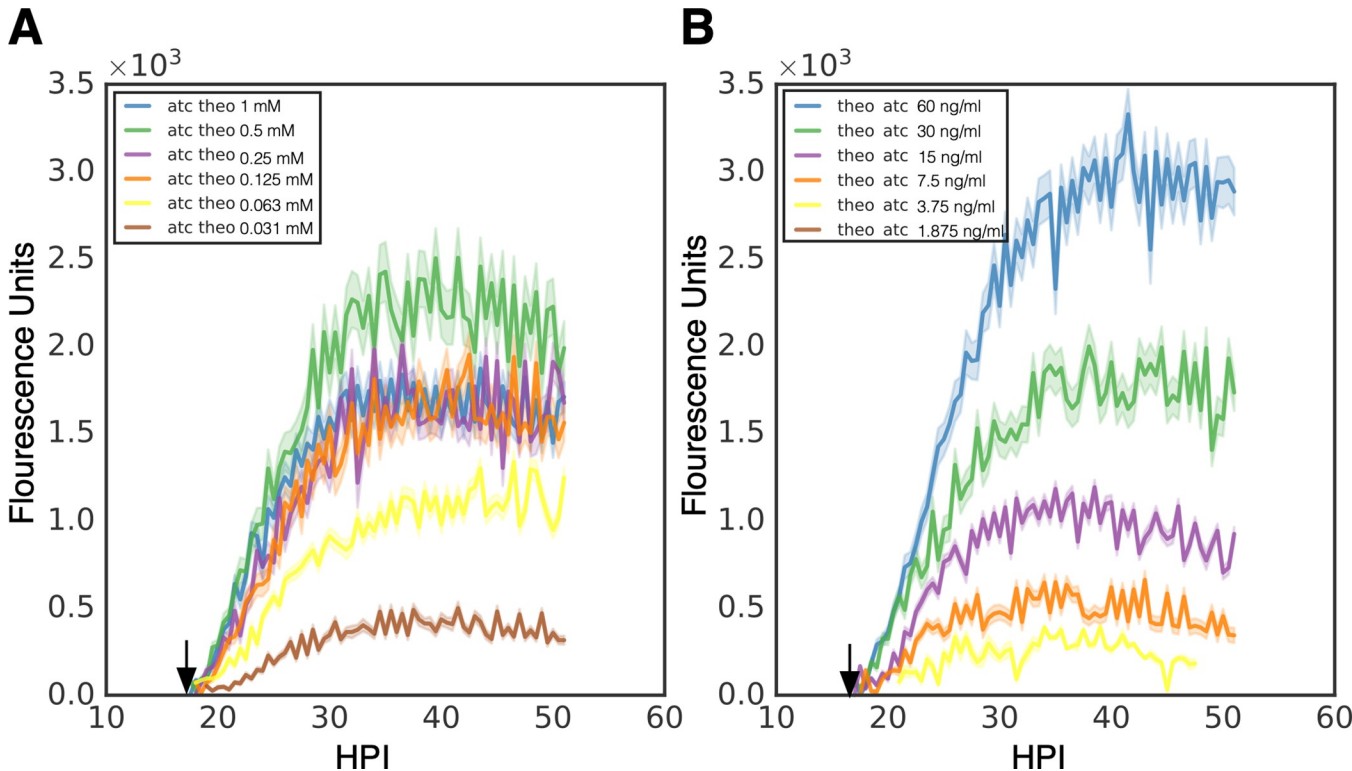

**Fig 4. Induction kinetics of p2TK2-SW2-Tet-J-E-clover-flag.** A) Cos-7 cells were infected with L2-Tet-J-E-clover-flag, treated with varying dilutions of theophylline at 16 hpi (1 mM, 0.5 mM, 0.25 mM, 0.125 mM, 0.0625 mM, 0.03125mM) and 30 ng/ml aTc at 0 hpi. The infections were monitored using live cell imaging for 50 hours. B) Cos-7 cells were infected with L2-Tet-riboJ-E-clover-flag, treated with varying dilutions of aTc at 16 hpi (60 ng/ml, 30 ng/ml, 15 ng/ml, 7.5 ng/ml, 3.75 ng/ml, 1.875 ng/m) and 0.5 mM theophylline at 0 hpi. The infections were monitored using live cell imaging for 50 hours. Expression intensities from >50 individual inclusions were monitored via automated live-cell fluorescence microscopy and the mean intensities are shown. Cloud represents SEM. Y-axes are denoted in scientific notation.

To test the effective repression of gene expression of this system p2TK2-SW2-Tet-J-E-hctB-3xflag (Fig 5A) was transformed into *Ctr* producing L2-Tet-J-E-hctB-flag. Unlike the p2TK2-SW2-E-hctB-3xflag construct, p2TK2-SW2-Tet-J-E-hctB-3xflag successfully transformed into *Ctr* without accumulating mutations suggesting tighter repression of leaky expression. Cos-7 cells were infected with L2-Tet-J-E-hctB-flag and induced for HctB expression with 30 ng/ml aTc and 0.5 mM theophylline at 16 hpi. Protein was isolated, separated by PAGE, blotted to nitrocellulose and expression was evaluated using an anti-flag antibody. HctB-flag was detected only when both inducers (aTc and theophylline) were used (Fig 5B). Gene expression was also assessed using confocal microscopy. Cos-7 cells were plated onto glass coverslips and infected with L2-Tet-J-E-hctB-flag. Gene expression was induced with 30 ng/ml aTc, 0.5 mM theophylline, or both at 16 hpi and the coverslips were fixed and stained with an anti-flag antibody at 30 hpi before mounting for confocal microscopy. Confocal microscopy confirmed HctB induction with both the transcription and translation inducer added (Fig 5C). However, HctB expression was detected at low levels when induced with aTc only suggesting translational repression with this construct was slightly leaky (Fig 5C).

As we could not transform p2TK2-SW2-T5-E-hctB-flag into *Chlamydia*, we hypothesized that expression of HctB inhibited the formation of the infectious EB cell form. To test this, Cos-7 cells were infected with L2-Tet-J-E-hctB-flag and induced with 30 ng/ml aTc, 0.5 mM theophylline, or both at 16 hpi and EBs were harvested at 30 hpi and 48 hpi. Induction of both transcription and translation resulted in a greater than 2.5 log reduction in infectious progeny

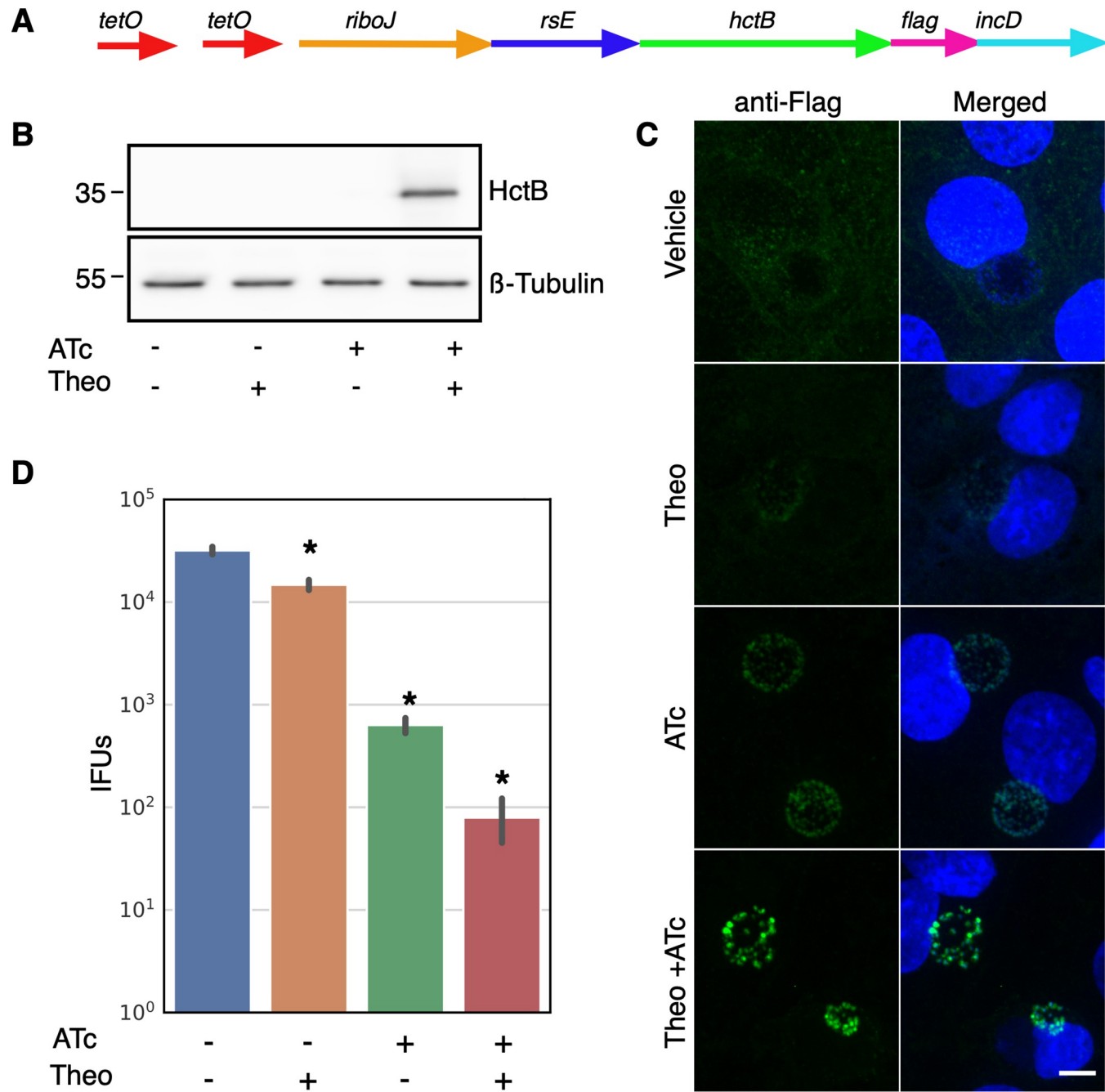

**Fig 5. Characterization of p2TK2-SW2-Tet-J-E-hctB-flag.** A) Schematic of the Te-riboJ-E-hctB-flag construct. HctB expression is controlled by an aTc inducible promoter, riboJ insulator and the riboE riboswitch. B) Anti-flag western blot of Cos-7 cells infected with L2-Tet-J-E-hctB-flag comparing expression of theophylline treated and untreated cultures. Cells were induced with 0.5 mM theophylline, 30ng/ml aTc, both aTc and theophylline or vehicle only at 16 hpi and proteins were harvested at 30 hpi. HctB-flag expression was only detected in the samples induced with both aTc and theophylline. C) Confocal micrographs of Cos-7 cells infected with L2-Tet-riboJ-E-hctB-flag, induced with 0.5 mM theophylline, 30ng/ml aTc, both aTc and theophylline or vehicle only at 16 hpi and fixed and stained with DAPI (blue) for confocal microscopy at 30 hpi. The flag tag was stained with a primary antibody to the flag and an alexa 488 anti-mouse secondary antibody (green). Size bar = 10 μm. D) Production of infectious progeny was determined using a reinfection assay. Cos-7 cells were infected with L2-Tet-J-E-hctB-flag and the production of infectious progeny was determined at 48 hpi after induction with 0.5 mM theophylline, 30ng/ml aTc, both aTc and theophylline or vehicle only at 16 hpi. Asterisks denote p-values < 0.05. Error bars = SEM.

at both 30 hpi and 48hpi (Fig 5D, S1 Fig). Confocal microscopy indicated that transcription induction with aTc only resulted in low but detectable HctB production (Fig 5C) and this leaky expression was also evident when assaying for infectious progeny. Induction of transcription only resulted in about a log reduction in infectious progeny at both 30 hpi and 48 hpi (Fig 5D, S1 Fig). Translation induction only resulted in a slight but statistically significant decrease in EB production as compared to no induction control (Fig 5D, S1 Fig). Together these data suggest that the combination of transcriptional repression and translational inducible regulation was significantly tight enabling *Ctr* to be successfully transformed with the construct and that induction was sufficiently high to induce the inhibition of the production of infectious progeny.

## Expression from T5 and Tet-J-E is cell type specific

Chlamydial infection of vertebrate cells consists of a multiple cell type developmental cycle. For *Ctr* L2 the elementary body (EB) cell type mediates cell entry and differentiates into the reticulate body (RB) cell type over an ~10 hour period before initiating cell division. The RB cell type undergoes growth and division leading to an expansion of RB numbers. The RB cell type also matures during this process eventually producing an intermediate body (IB) cell type that matures back into the EB cell form over ~8 hours [19]. Our studies have shown that different promoters are active in these distinct cell populations [19]. Confocal microscopy of Clover expression and flag staining for both the T5-E-clover-flag and Tet-J-E-clover-flag constructs appeared non uniform in the inclusion suggesting expression in only a subset of cells. To determine the cells in which these promoters were active we replaced Clover in both of these constructs with the GFP variant Neongreen and added an inframe LVA degradation tag to produce the plasmids p2TK2-SW2-T5-E-ngLVA-3xFlag and p2TK2-SW2-Tet-J-E-ngLVA-3xFlag. Neongreen-LVA (ngLVA) protein had a halflife of ~30 minutes in *Ctr* (S3 Fig). The plasmids were transformed into *Ctr* and the expression pattern of these constructs was compared to that of p2TK2-SW2-euoprom-ngLVA. p2TK2-SW2-euoprom-ngLVA, like p2TK2-SW2-euoprom-Clover [19] uses the euo promoter to drive expression specifically in the RB cell type. Cos-7 cells infected with L2 T5-E-ngLVA, L2 Tet-J-E-ngLVA, or L2 euoprom-ngLVA were fixed for confocal microscopy at 30 hpi. Expression of ngLVA for all three promoters was very similar showing expression in a subset of large cells suggestive of RBs (Fig 6A). Quantification of the number of cells per inclusion that expressed ngLVA from each promoter showed that all three promoters expressed ngLVA in similar numbers of cells (Fig 6B). This suggests that both of the synthetic sigma70 optimized promoters (T5 and Tet) when used in *Ctr* expressed primarily in the RB cell type and not in the IB.

## Conclusions

Ectopic gene expression is an important tool for uncovering the function of potential virulence associated genes in pathogenic bacteria. We have adapted the E riboswitch, a theophylline binding aptamer, to regulate gene translation in *Ctr*. Riboswitches have been used in many organisms to regulate gene expression [16, 24, 25, 37–39]. In bacteria, riboswitches are constructed of aptamers that fold to block ribosome assembly at the translational start site in the absence of their cognate ligand. This translational control can be combined with strong synthetic promoters, native promoters, cell type specific promoters, temporal promoters or inducible promoters to add increasingly granular expression control of effectors and regulatory proteins. In this study we combined the E riboswitch with the strong synthetic promoter T5 and demonstrated that the riboswitch efficiently repressed translation of Clover and was strongly inducible by theophylline. In addition this induction was dose responsive providing

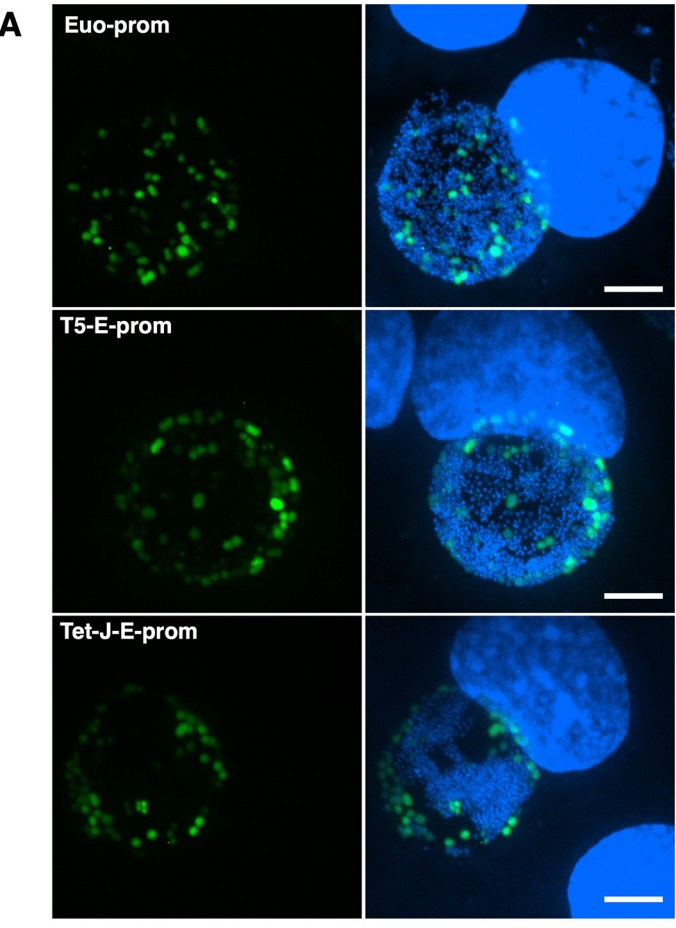

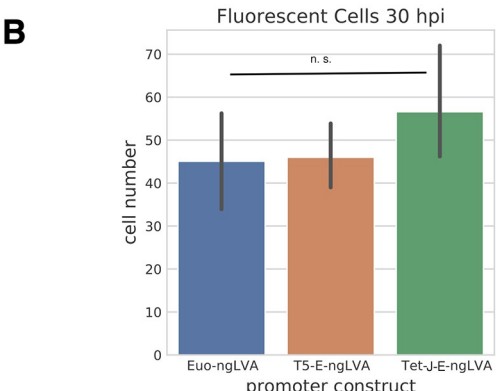

**Fig 6. Promoter cell type expression.** A) Confocal micrographs of Cos-7 cells infected with L2-euo-neogreenLVA (euo-ngLVA), L2-T5-E-neogreenLVA (T5-E-ngLVA), or L2-Tet-J-E-neogreenLVA (Tet-E-ngLVA) (green) and fixed and stained with DAPI (blue) at 30 hpi. Size bar = 10 μm. B) Quantification of > 20 neongreenLVA expressing chlamydial cells for each promoter construct. Error bars = SEM. n. s. denotes p-values > 0.05.

an excellent tool for the control of ectopic gene expression. We recently published the use of the T5prom-E riboswitch for the ectopic expression of three chlamydial proteins, dbdA, hctA, and scc1 demonstrating its utility in dissecting the function of chlamydial proteins [17].

In addition to combining translational expression control with a strong promoter, we also demonstrated that it can be used with a native chlamydial promoter. The E riboswitch was cloned upstream of the ORF for the plasmid gene *pgp4*. Pgp4 is a gene expression regulator for both chromosomal and plasmids genes and the absence of *pgp4* results in a loss of glycogen accumulation in the inclusion [33]. The addition of the E riboswitch led to undetectable levels of Pgp4 expression and the loss of glycogen accumulation when theophylline was absent. The addition of theophylline during infection restored functional levels of Pgp4 as demonstrated by the restoration of glycogen accumulation in the inclusion and detectable expression of Pgp4 via western blot and confocal microscopy.

By combining translational control with transcriptional control we were able to improve the repression of protein expression. The T5 promoter-E-riboswitch combination proved to have undetectable expression when driving Clover expression as assessed by western blotting and confocal microscopy. However, when attempting to express the chlamydial protein HctB, a protein involved in controlling the developmental cycle, leaky expression resulted in mutation of the plasmid causing HctB to not express. It is interesting to note that ectopic expression of HctA using just the T5prom-E system transformed into chlamydia without mutations suggests low levels of HctA may be more tolerated.

By combining translational control (E riboswitch) with transcription control (Tet inducible promoter), we created an extremely tightly regulated gene expression system. The E riboswitch requires the 5'-UTR of the transcript to properly fold and block the ribosome binding site of the transcript. Combining the E riboswitch with different promoters and different transcription start sites can potentially affect the folding of the riboswitch, thus changing its repression and induction properties. To eliminate this effect and increase the reliability of the riboswitch in relation to a variety of promoters, we cloned the riboJ ribozyme insulator upstream of the E riboswitch. The riboJ insulator is made up of the sTRSV-ribozyme with an additional 23-nt hairpin immediately downstream [35, 40, 41]. This hairpin imposes structure to the UTR just upstream of the E riboswitch, minimizing its influence on the folding of the riboswitch and ensuring any upstream structure is consistent between promoters.

Surprisingly, the order of induction (transcription vs translation) did not significantly change the gene expression kinetics suggesting that there was not an accumulation of transcripts after Tet induction that then could be induced to initiate translation. Instead, this observation suggests the transcripts either don't accumulate or, after folding into the inhibited structure in the absence of theophylline they don't then refold revealing the RBS upon theophylline addition. This suggests theophylline binding competes with inhibitory folding during mRNA synthesis.

HctB, when cloned into this dual induction plasmid was successfully transformed into *Ctr* and was inducible with the addition of both theophylline and aTc as detected by western blotting and confocal microscopy. Additionally, ectopic expression of HctB early in infection (16 hpi) significantly reduced the formation of infectious progeny. Together, these data confirm that leaky expression from T5-E likely rendered successful transformation of this clone impossible. Therefore, the combination of transcriptional and translational control is an ideal system to study the effects of toxic proteins or proteins that regulate the developmental cycle.

Interestingly, both the T5-E and Tet-riboJ-E promoter systems appear to only significantly express in the RB cell type. The promoters for both of these constructs are based on *E. coli* sigma70 consensus sequences and are constitutive in many bacteria [42, 43]. In *Ctr* these promoters appear to express primarily in the RB cell type suggesting gene expression in the intermediate body (IB) and EB cell type may require specific promoters or additional regulatory elements. Our data suggest that the ability to add translational control independently from

transcriptional control using riboJ ribozyme and E riboswitch will be an important tool in controlling ectopic gene expression in these chlamydial cell types.

Adding inducible translational control to the tool box for chlamydial genetic tools increases opportunities to unveil the function of *Ctr* regulatory genes and effector genes to reveal their role in pathogenesis. The ability to control the timing and strength of gene expression independently from promoter strength and timing increases the utility of ectopic gene expression and provides an important tool for studying chlamydial pathogenesis.

## Supporting information

**S1 Fig. IFU results at 30 hpi.** A) Cos-7 cells were infected with L2-T5-E-clover-flag and the production of infectious progeny was determined at 30 hpi after theophylline induction or vehicle only. B) Cos-7 cells were infected with L2-nprom-E-pgp4-flag and the production of infectious progeny was determined at 30 hpi after theophylline induction or vehicle only. C) Cos-7 cells were infected with L2-Tet-J-E-clover-flag and the production of infectious progeny was determined at 30 hpi after induction with 0.5 mM theophylline, 30ng/ml aTc, both aTc and theophylline or vehicle only at 16 hpi. D) Cos-7 cells were infected with L2-Tet-J-E-hctB-flag and the production of infectious progeny was determined at 30 hpi after induction with 0.5 mM theophylline, 30ng/ml aTc, both aTc and theophylline or vehicle only at 16 hpi. Asterisks denote p-values < 0.05. Error bars = SEM.
(TIF)

**S2 Fig. RNA-seq analysis of p2TK2-SW2-Tetprom-riboJ-E-hctB-flag.** Cos-7 cells infected with L2-Tet-J-E-hctB-flag were induced with 0.5 mM theophylline and 30ng/ml aTc at 15 hpi and RNA was harvested at 24 hpi. RNA was processed for next-gen RNA-seq sequencing. Aligned reads are shown with the schematic of the Tet-J-E-hctB.
(TIF)

**S3 Fig. ngLVA degradation kinetics.** Cos-7 cell infected with L2-euoprom-ngLVA, treated with 34 μg/ml chloramphenicol (CAM) or vehicle only (1:1000 EtOH in RPMI) at 35 hpi (arrow). The infections were monitored using live cell imaging for 50 hours. Expression intensities from >50 individual inclusions were measured via automated live-cell fluorescence microscopy and the mean intensities are shown. Cloud represents SEM. Y-axes are denoted in scientific notation. Chloramphenicol treated sample showed a decrease of half max intensity 30 mins after treatment.
(TIFF)

**S4 Fig. Uncropped western blots.**
(PDF)

**S1 Table. Primers and templates used for plasmid construction.**
(PDF)

## Acknowledgments

We would like to thank Dr. Paul Beare at the Rocky Mountain Labs for providing us the Tet repressor and promoter sequence. We would also like to thanks Dr. John-Demian (JD) Sauer for providing us with the T5prom-E promoter and riboswitch.

## Author Contributions

**Conceptualization:** Nicole A. Grieshaber, Scott S. Grieshaber.

**Data curation:** Scott S. Grieshaber.

**Formal analysis:** Nicole A. Grieshaber, Scott S. Grieshaber.

**Funding acquisition:** Nicole A. Grieshaber, Scott S. Grieshaber.

**Investigation:** Nicole A. Grieshaber, Travis J. Chiarelli, Cody R. Appa, Grace Neiswanger, Kristina Peretti, Scott S. Grieshaber.

**Supervision:** Nicole A. Grieshaber, Scott S. Grieshaber.

**Writing – original draft:** Nicole A. Grieshaber, Scott S. Grieshaber.

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
