## [Decision Letter · Decision Letter 0]

5 Oct 2021

PONE-D-21-27589Translational gene expression control in Chlamydia trachomatis.PLOS ONE

Dear Dr. Grieshaber,

Thank you for submitting your manuscript to PLOS ONE. After careful consideration, we feel that it has merit but does not fully meet PLOS ONE’s publication criteria as it currently stands. Therefore, we invite you to submit a revised version of the manuscript that addresses the points raised during the review process.

We look forward to receiving your revised manuscript.

Kind regards,

Jorn Coers

Academic Editor

PLOS ONE

Journal Requirements:

Additional Editor Comments:

Both reviewers asked for a Chlamydia loading control. These concerns need to be addressed with the relevant data provided in a revised MS

Reviewers' comments:

Reviewer's Responses to Questions

**Comments to the Author**

1. Is the manuscript technically sound, and do the data support the conclusions?

Reviewer #1: Partly

Reviewer #2: Partly

2. Has the statistical analysis been performed appropriately and rigorously? 

Reviewer #1: Yes

Reviewer #2: Yes

3. Have the authors made all data underlying the findings in their manuscript fully available?

Reviewer #1: Yes

Reviewer #2: No

4. Is the manuscript presented in an intelligible fashion and written in standard English?

Reviewer #1: Yes

Reviewer #2: Yes

5. Review Comments to the Author

Reviewer #1: In this manuscript by Grieshaber et al., the authors describe a new method of translational control, riboswitch E, for use in Chlamydia trachomatis. The ability to control translation of chlamydial factors will enable functional characterization of regulatory factors and effectors. Overall the manuscript is well written and easy to follow, however I have some concerns related to data presentation.

Major comments:

Fig 1B, Fig. 3B, Fig. 4B, etc.: A loading control (Ctr HSP) should be included.

In my opinion, Fig. 2A should be part of Fig. 1 (maybe Fig. 1D?). The bar graph should also have N.S. to indicate no significant difference following Theo treatment.

The decimal seems to be missing in the legend in Fig. 2B. Should also include mM in the legend.

Fig. 7 should have N.S. on graph to indicate no significant difference.

Scale bars are missing from many of the microscopy images.

Details on the number of experimental replicates and/or technical replicates is missing. This should be included in the figure legends or materials and methods section.

Minor comments:

In many places, Chlamydia is italicized but trachomatis is not. Both should be italicized.

In some instances, the authors use Chlamydia trachomatis whereas in others they use Ctr. Consistent use of one or the other should be used.

Line 237, 454 should be 434

Reviewer #2: Greishaber et al. describe the development of a theophylline-inducible riboswitch E that allows for regulatable translation in Chlamydia trachomatis L2 from plasmid-encoded genes. Riboswitch-regulation was assessed with both native (pgp4) and synthetic promoters (T5-lac and tet). The tet-promoter system allowed for transcriptional control (tet operator/repressor) in addition to translational control via the riboswitch. In general, the data are convincing and regulation appears to be very tight and dose-responsive. The riboswitch approach should have significant utility for the chlamydial field as highlighted by the construction of hctB-plasmid transformants using the dual tet-riboswitch E vector to negate HctB-toxicity owing to leaky expression. In addition, the authors document that the T5-lac/tet promoters appear to show developmental-stage specific expression which has relevance for many of the chlamydial expression-vectors in use. Suggestions for improving the manuscript are listed below as major and minor criticisms.

Major criticisms

1) The Flag-tag western blots showing induction of the protein of interest in figures 1, 3, 4, and 6 would greatly benefit from loading controls to demonstrate that chlamydial proteins were present in the Flag-tag negative lanes. Inclusion of sample-matched anti-MOMP western blots (or another prominent chlamydial protein) would address this weakness.

2) The group has already reported the use of the riboswitch E in a recent publication. The study is referenced in the methods regarding how transformations were performed (ref 38, line 132), but nowhere else. It is clear that the current manuscript is an in-depth analysis of the riboswitch and it provides significant new details that merit sharing with the field. A brief discussion of the use of the riboswitch in the prior study would further support the utility of the system and it is recommended that the authors highlight the prior usage with dbdA, hctA, and scc1.

Minor criticisms

1) The RNAseq data under SRA 10220676 (https://www.ncbi.nlm.nih.gov/sra/?term=10220676, accessed on 9-30-21) do not appear to match the current study. Please confirm the submission number.

2) The methods mention using pen G or spec as appropriate for the plasmids (lines 133-135). All of the plasmids reported appear to be p2TK2-SW2 based (ref 20), which uses the bla marker. Please indicate which vector required spec-selection.

3) Please state in the methods the vehicle used to dissolve theophylline.

4) Line 265 – “two” should be too

5) The methods state that glycogen staining was performed at 40 hpi (line 197), while the results indicate 36 hpi (328). Please clarify.

6) When discussing the toxicity of HctB leakiness in the discussion (lines 586-593) it is perhaps worth noting that HctA (ref 38) did not require a dual transcription/translation regulated system to obtain transformants. Is this due to less leakiness for the hctA riboswitch construct or differences in the impact of these proteins on development?

7) Lines 588-589: “Additionally, ectopic expression of HctB early in infection (16 hpi) inhibited the formation of infectious progeny.” Recommend replacing “inhibited” with significantly reduced. At least in my mind, inhibited implies a complete block in IFU production which is not the case (figure 6D).

8) Recommend unifying the Y-axis formats in Fig_S1.

6. PLOS authors have the option to publish the peer review history of their article (what does this mean?). If published, this will include your full peer review and any attached files.

Reviewer #1: No

Reviewer #2: No

---

## [Author Response · Author response to Decision Letter 0]

8 Nov 2021

We have addressed the reviews comments as follows:

Both reviewers asked for a Chlamydia loading control. These concerns need to be addressed with the relevant data provided in a revised MS

We have provided a tubulin control as loading controls for the westerns. We chose to not use a chlamydial antigen as induction of both Pgp4 and HctB affect the developmental cycle of chlamydial making any chlamydial protein a poor choice for a loading control.

Comments to the Author

Reviewer #1: In this manuscript by Grieshaber et al., the authors describe a new method of translational control, riboswitch E, for use in Chlamydia trachomatis. The ability to control translation of chlamydial factors will enable functional characterization of regulatory factors and effectors. Overall the manuscript is well written and easy to follow, however I have some concerns related to data presentation.

Major comments:

Fig 1B, Fig. 3B, Fig. 4B, etc.: A loading control (Ctr HSP) should be included.

We have added beta tubulin control as loading controls for the westerns. We chose not to use a chlamydial antigen as induction of both Pgp4 and HctB affect the developmental cycle of chlamydial making any chlamydial protein a poor choice for a loading control.

In my opinion, Fig. 2A should be part of Fig. 1 (maybe Fig. 1D?). The bar graph should also have N.S. to indicate no significant difference following Theo treatment.

We agreed and updated the figure and text. We have added N. S. Where appropriate

The decimal seems to be missing in the legend in Fig. 2B. Should also include mM in the legend.

Fixed

Fig. 7 should have N.S. on graphs to indicate no significant difference.

This is now addressed

Scale bars are missing from many of the microscopy images.

This is now addressed

Details on the number of experimental replicates and/or technical replicates is missing. This should be included in the figure legends or materials and methods section.

This is now addressed

Minor comments:

In many places, Chlamydia is italicized but trachomatis is not. Both should be italicized.

This is now addressed

In some instances, the authors use Chlamydia trachomatis whereas in others they use Ctr. Consistent use of one or the other should be used.

We are now consistently using Ctr

Line 237, 454 should be 434

fixed

Reviewer #2: Greishaber et al. describe the development of a theophylline-inducible riboswitch E that allows for regulatable translation in Chlamydia trachomatis L2 from plasmid-encoded genes. Riboswitch-regulation was assessed with both native (pgp4) and synthetic promoters (T5-lac and tet). The tet-promoter system allowed for transcriptional control (tet operator/repressor) in addition to translational control via the riboswitch. In general, the data are convincing and regulation appears to be very tight and dose-responsive. The riboswitch approach should have significant utility for the chlamydial field as highlighted by the construction of hctB-plasmid transformants using the dual tet-riboswitch E vector to negate HctB-toxicity owing to leaky expression. In addition, the authors document that the T5-lac/tet promoters appear to show developmental-stage specific expression which has relevance for many of the chlamydial expression-vectors in use. Suggestions for improving the manuscript are listed below as major and minor criticisms.

Major criticisms

1) The Flag-tag western blots showing induction of the protein of interest in figures 1, 3, 4, and 6 would greatly benefit from loading controls to demonstrate that chlamydial proteins were present in the Flag-tag negative lanes. Inclusion of sample-matched anti-MOMP western blots (or another prominent chlamydial protein) would address this weakness.

We have added a beta tubulin control as loading controls for the westerns. We chose not to use a chlamydial antigen as induction of both Pgp4 and HctB affect the developmental cycle of chlamydial making any chlamydial protein a poor choice for a loading control.

2) The group has already reported the use of the riboswitch E in a recent publication. The study is referenced in the methods regarding how transformations were performed (ref 38, line 132), but nowhere else. It is clear that the current manuscript is an in-depth analysis of the riboswitch and it provides significant new details that merit sharing with the field. A brief discussion of the use of the riboswitch in the prior study would further support the utility of the system and it is recommended that the authors highlight the prior usage with dbdA, hctA, and scc1.

We have added this to the discussion

Minor criticisms

1) The RNAseq data under SRA 10220676 (https://www.ncbi.nlm.nih.gov/sra/?term=10220676, accessed on 9-30-21) do not appear to match the current study. Please confirm the submission number.

Our mistake, this was the temporary accession number the correct one is PRJNA756251

The methods mention using pen G or spec as appropriate for the plasmids (lines 133-135). All of the plasmids reported appear to be p2TK2-SW2 based (ref 20), which uses the bla marker. Please indicate which vector required spec-selection.

We have updated the material and methods to reflect that we replaced bla with spec in this construct

3) Please state in the methods the vehicle used to dissolve theophylline.

Theophylline was dissolved straight into RPMI media. We have added this to the materials and methods

4) Line 265 – “two” should be too

Fixed

5) The methods state that glycogen staining was performed at 40 hpi (line 197), while the results indicate 36 hpi (328). Please clarify.

This was done at 36 hours and has been updated in the materials and methods

6) When discussing the toxicity of HctB leakiness in the discussion (lines 586-593) it is perhaps worth noting that HctA (ref 38) did not require a dual transcription/translation regulated system to obtain transformants. Is this due to less leakiness for the hctA riboswitch construct or differences in the impact of these proteins on development?

We elaborate on this in the discussion. Basically HctA does not appear to be as toxic even though there is some evidence of leakiness.

7) Lines 588-589: “Additionally, ectopic expression of HctB early in infection (16 hpi) inhibited the formation of infectious progeny.” Recommend replacing “inhibited” with significantly reduced. At least in my mind, inhibited implies a complete block in IFU production which is not the case (figure 6D).

We agree and have updated the sentence. 

8) Recommend unifying the Y-axis formats in Fig_S1.

Agreed and fixed

---

## [Decision Letter · Decision Letter 1]

17 Nov 2021

PONE-D-21-27589R1Translational gene expression control in Chlamydia trachomatisPLOS ONE

Dear Dr. Grieshaber,

There are some minor points that need to be addressed (missing file for supplementary figure S2, etc.).  Please , send in revised version that addresses these minor point and I will make an editorial decision shortly thereafter. Thanks!

We look forward to receiving your revised manuscript.

Kind regards,

Jorn Coers

Academic Editor

PLOS ONE

Journal Requirements:

Reviewers' comments:

Reviewer's Responses to Questions

**Comments to the Author**

1. If the authors have adequately addressed your comments raised in a previous round of review and you feel that this manuscript is now acceptable for publication, you may indicate that here to bypass the “Comments to the Author” section, enter your conflict of interest statement in the “Confidential to Editor” section, and submit your "Accept" recommendation.

Reviewer #1: All comments have been addressed

Reviewer #2: (No Response)

2. Is the manuscript technically sound, and do the data support the conclusions?

Reviewer #1: Yes

Reviewer #2: Yes

3. Has the statistical analysis been performed appropriately and rigorously? 

Reviewer #1: Yes

Reviewer #2: Yes

4. Have the authors made all data underlying the findings in their manuscript fully available?

Reviewer #1: Yes

Reviewer #2: Yes

5. Is the manuscript presented in an intelligible fashion and written in standard English?

Reviewer #1: Yes

Reviewer #2: Yes

6. Review Comments to the Author

Reviewer #1: (No Response)

Reviewer #2: The authors did an excellent job responding to the prior criticisms. Two minor points that should be addressed:

1) A file for Figure S2 (RNA-seq results) does not appear to be included with the revised submission. Page 35 is S1 and page 36 is S3. If the figure S2 intended for the revised submission matches the first submission, then I do not have additional comments for this data.

2) The authors indicate that they revised Figure S1 to unify the Y-axis formats in response to the round 1 reviewer comment. The included Figure S1 still uses 3 different numbering formats. Perhaps the incorrect version of the figure was uploaded. Recommend using scientific notation for panels A and B as shown in panels C and D.

7. PLOS authors have the option to publish the peer review history of their article (what does this mean?). If published, this will include your full peer review and any attached files.

Reviewer #1: No

Reviewer #2: No

---

## [Author Response · Author response to Decision Letter 1]

7 Dec 2021

We have addressed the reviewers comments as follows: 

Second review

Reviewer #2: The authors did an excellent job responding to the prior criticisms. Two minor points that should be addressed:

1) A file for Figure S2 (RNA-seq results) does not appear to be included with the revised submission. Page 35 is S1 and page 36 is S3. If the figure S2 intended for the revised submission matches the first submission, then I do not have additional comments for this data.

Yes this was an oversight on resubmission. Fig S2 is unchanged from the original submission and is now included in this submission.

2) The authors indicate that they revised Figure S1 to unify the Y-axis formats in response to the round 1 reviewer comment. The included Figure S1 still uses 3 different numbering formats. Perhaps the incorrect version of the figure was uploaded. Recommend using scientific notation for panels A and B as shown in panels C and D.

We have fixed the graphs in fig S1 to all have the same style of Y axis.

---

## [Editor Report · Decision Letter 2]

13 Jan 2022

Translational gene expression control in Chlamydia trachomatis

PONE-D-21-27589R2

Dear Dr. Grieshaber,

We’re pleased to inform you that your manuscript has been judged scientifically suitable for publication and will be formally accepted for publication once it meets all outstanding technical requirements.

Kind regards,

Jorn Coers

Academic Editor

PLOS ONE

Additional Editor Comments (optional):

the second revision address the very minor outstanding critique points. Congratulations to having your work accepted for publication.
---

## [Editor Report · Acceptance letter]

18 Jan 2022

PONE-D-21-27589R2 

Translational gene expression control in *Chlamydia trachomatis*. 

Dear Dr. Grieshaber:

I'm pleased to inform you that your manuscript has been deemed suitable for publication in PLOS ONE. Congratulations! Your manuscript is now with our production department. 

Kind regards, 

on behalf of

Dr. Jorn Coers 

Academic Editor

PLOS ONE